# Balanced Mixture of SuperNets
# for Learning the CNN Pooling Architecture

**Mehraveh Javan Roshtkhari**[1]  **Matthew Toews**[1]  **Marco Pedersoli**[1]

[1]École de technologie supérieure (ÉTS), Montreal, Canada

**Abstract**   Downsampling layers, including pooling and strided convolutions, are crucial components of the convolutional neural network architecture that determine both the granularity/scale of image feature analysis as well as the receptive field size of a given layer. To fully understand this problem, we analyse the performance of models independently trained with each pooling configurations on CIFAR10, using a ResNet20 network and show that the position of the downsampling layers can highly influence the performance of a network and predefined downsampling configurations are not optimal.

Network Architecture Search (NAS) might be used to optimize downsampling configurations as an hyperparameter. However, we find that common one-shot NAS based on a single SuperNet does not work for this problem. We argue that this is because a SuperNet trained for finding the optimal pooling configuration fully shares its parameters among all pooling configurations. This makes its training hard because learning some configurations can harm the performance of others.

Therefore, we propose a balanced mixture of SuperNets that automatically associates pooling configurations to different weight models and helps to reduce the weight-sharing and inter-influence of pooling configurations on the SuperNet parameters. We evaluate our proposed approach on CIFAR10, CIFAR100, as well as Food101, and show that in all cases our model outperforms other approaches and improves over the default pooling configurations.

## 1 Introduction

Downsampling layers in convolutional neural networks (CNN) are crucial, as they provide robustness to shift and scale variations (Zhao et al., 2017), reduce the computational cost of models (Jin et al., 2021; Riad et al., 2022), and control the access of subsequent convolution kernels to spatial information, determining their receptive field (Le and Borji, 2017; Luo et al., 2016). In CNNs, spatial resolution is related to the receptive field, which determines the aggregation of local features and affects the performance of the CNN (Jang et al., 2022; Richter and Pal, 2022). The receptive field in turn is controlled indirectly by the hyperparametrs of the network such as depth, filter sizes and downsampling/pooling layers. The spatial density of the content in a dataset highly affects the optimal receptive field and therefore the spatial pooling configuration. For instance, for a good recognition on textures, smaller and more detailed local patterns are more important (Jang et al., 2022), while for shapes, considering larger regions of the image should provide a better representation (Luo et al., 2016). Thus, being able to select how to downsample the image representation in CNNs can help to better adapt the representation to the specific characteristics of a given dataset and help to better understand the way that convolutional neural network find meaningful patterns in images, and therefore determine what are the relevant features for a given task (Riad et al., 2022).

In CNN design, feature map downsampling is commonly performed by applying a strided convolution (Howard et al., 2017), a convolution followed by a pooling operation (Simonyan and Zisserman, 2014; Szegedy et al., 2015) or a combination of the two (He et al., 2016). For a downstream task such as classification, the position of the downsampling in a network architecture is pre-defined

and based on the assumption that the receptive field should increase over layers until covering most of the image (Richter and Pal, 2022). While this assumption can be removed by the use of self-attention (Chen et al., 2021), its usage seems still utterly important for a good trade-off of computation and accuracy (Liu et al., 2021). In this work we show that the commonly used pooling configurations may not be optimal. A possible solution is to learn the best pooling configuration for the dataset at hand. However, pooling configurations are discrete parameters and the number of candidate architectures grows exponentially with depth, making bruteforcefully searching for the best pooling configuration computationally infeasible for modern CNNs.

Previous works that attempt to find optimal feature map sizes in a predefined architecture avoid the discrete nature of sub-sampling layers by relaxing the problem by learning resizing modules (Liu et al., 2020; Riad et al., 2022; Jang et al., 2022), or indirectly do so by learning continuous filter sizes (Romero et al., 2021; Pintea et al., 2021; Romero et al., 2021) at the same time as training the CNN. Some works, such as DiffStride (Riad et al., 2022), cast learning fractional strides as learning cropping size in frequency domain, the pooling is performed in spectral domain resulting in higher cost and involving complex value operations.

Differently than previous work, we cast the problem of finding the optimal scales of analyzing the CNN features as a Neural Architecture Search (NAS) problem. A popular research direction for solving NAS problem is to first relax the optimization problem into an equivalent, but differentiable one and then find the optimal hyper-parameters through bi-level optimization (Liu et al., 2018b). The search is then reduced to the training of a single over-parameterized network that contains all the searchable configurations, commonly called a SuperNet. However, differentiable models are computationally and memory-wise demanding, because they evaluate all model configurations at each training iteration. Additionally, they do not always provide optimal solutions as the bi-level optimization is heavily approximated (Xue et al., 2021) and it is difficult to impose constraints in configurations (as some configurations might not be feasible).

An alternative is to train a SuperNet by sampling at each iteration a different sub-net (Guo et al., 2020; Li and Talwalkar, 2020; Stamoulis et al., 2020), while selecting the most likely sub-net (Single Path Single-Shot. This solves the problems of computation and memory and is not limited to differentiable models. However, finding the optimal subnet during training is quite risky because the estimation of the gradients for sampling based algorithms is very noisy (Li and Talwalkar, 2020) and the learning can easily be misled by this noise. These NAS approaches are sometimes referred to as coupled or one-stage approaches, since the training of the SuperNet and searching for the optimal configuration are performed together. In general, coupled optimization of architecture and weights suffer from bias towards rapidly converging networks and multi-model forgetting (He et al., 2021). A promising alternative is a two-stage search method, where a SuperNet is used to sample configurations uniformly with shared parameters for training, but no specific configuration is selected or preferred (Guo et al., 2020). Instead, after training, at search stage, the best configurations are evaluated on a validation set and selected. Even though in this case, training might be slightly longer, because it does not favor any sub-net, the simplicity and the robustness of the provided results make it a promising candidate for NAS, especially for those problems that are difficult to be relaxed in a differentiable way (Ren et al., 2021). For a more detailed analysis of related work see Appendix A.

In this work we tested both differentiable and sample based approaches, but both failed to provide good results for finding the optimal pooling configuration of a network. We hypothesize that the underlying reason is two fold: inappropriate search space design, and strong weight sharing in SuperNet. We show that defining the search space naively, by treating each resolution similar to independent operations, does not necessarily return better results than fully sharing weights among all resolutions, despite lower degree of weight sharing among the former. This search space design result in greedily reducing the weight sharing, i.e. all configurations with the same resolution at a

layer share weights, regardless of the path as a whole. Therefore, even though complete weight sharing pose a problem, its reduction should be performed in a more appropriate way.

To investigate this problem, we perform extensive experiments on on CIFAR10 dataset to find the optimal pooling configuration on ResNet20. As the resolution of CIFAR10 images is low, ResNet20 requires only 2 pooling layers, which amounts (after reasonable constraints) to 36 configurations. Thus, we have independently trained all configurations and consider the obtained accuracy as our ground truth performance. A revealing result is that, even with only 36 possible configurations and the extreme use of CIFAR10 for image classification, the standard pooling strategy is not the optimal and there is a gap of more than one point.

In order to find the optimal pooling configuration, we propose a new model based on SuperNet sampling that reduces the problem of parameter sharing by using multiple balanced SuperNets. The sampling of the pooling configuration is kept uniform as in (Guo et al., 2020), to avoid to introducing bias. However, to avoid interference among the different pooling configurations, we train multiple models at the same time. Each configuration favors sampling the model that leads to higher accuracy with it, while making sure that all models on average receive equal amounts of training, so that they are balanced. This training strategy allows each model to specialize to different pooling configurations.

Our main contributions are summarized as follows:

- We present the task of finding the optimal CNN downsampling or pooling layers as a NAS problem, and perform extensive experiments to evaluate search space design and NAS methods on the CIFAR10 classification task with ResNet20 architecture. We show that designing the search space for this problem requires more insight and a naive design can lead to poor pooling configurations.

- We show that, while optimal pooling configurations can improve upon the performance of standard configurations on the widely used CIFAR10 dataset, they are not identified by common NAS methods. We argue that this is due to weight sharing in the SuperNet and more specifically to the full weight sharing of the problem.

- We propose a balanced mixture of SuperNets that reduces the weight sharing problem by learning the correct association between each pooling configuration and one of the weight models.

- We validate our approach on several datasets and CNN configurations and show that by only learning the optimal scales with our method, we can improve the classification performance of ResNet architectures without altering any other hyperparameters.

## 2 Our Approach: Balanced Mixture of SuperNets

In this section we present the search space that we use in order to find the optimal pooling configurations. Then we present the corresponding SuperNet model and finally the balanced mixture of SuperNets we propose to tackle the full weight sharing problem.

### 2.1 Search Space

In this work we focus on finding the optimal spatial resolution of the feature maps in a CNN, that are controlled by downsampling operations.

First, we consider the general search space that contain $r$ resolutions per layer. The downsampling is performed by applying the most commonly used downsampling operations such as max pooling, reducing feature map size by a factor of two. Similarly to typical layer-wise operations, we can assign unique convolutional operations to each resolution at each layer. Considering $L$ layers, this will result in search space of size $r^L$. We note that for classification it is well known that the resolution of feature map is reduced across layers. Therefore, paths in the search space that contain upsampling operations are not appropriate for this task.

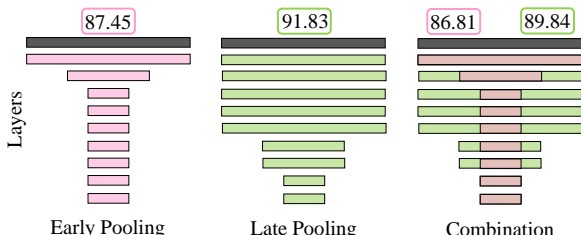

Figure 1: Interference of representations for ResNet20 on CIFAR10. *Early pooling* and *late pooling* produce different feature representations on their layers, which lead to different performance (on top). When training a model by sampling either one or the other pooling configuration (*combination*), the two representations interfere which leads to lower performance of both models. This motivated us to propose a mixture of models.

As downsampling operations reduce the feature map size, the boundary of this search space is determined by input size and the minimum feature map size expected before the classification layer. We consider the same number of downsampling operations as a default network (i.e. the predefined network configuration) and exclude from the search space the first pooling layer as the it corresponds to a manipulation of the input data. With these restrictions, the search space size is combination $\binom{L-1}{p}$, exponentially growing with the depth of the network. With $p + 1$ resolutions present at the search space, each architecture in this search space can be uniquely identified by the number of blocks in each resolution as $\alpha = [n_0, n_1, ..n_{p+1}]$, where $n_i$ is the number of blocks in resolution $i$ and $\sum_i n_i = L$. The search space design for each experiment is detailed in Appendix B.2 in tab. 5

For simplicity we use a pre-defined number of channels for all architectures, ensuring the same number of parameters in all architectures. We choose ResNet (He et al., 2016) as the building block of our search space as it is one of the most widely used and well studied architectures and ensured the incorporation of skip connections in our search space. As in ResNet the basic block is not a single convolutional layer but a block (with two convolutional layers and the skip connection), we use this block instead of a layer as basic unit to move the pooling location.

## 2.2 SuperNet

In order to find the optimal pooling configuration, we follow a two-stage strategy in which we first sample all configurations during training and then evaluate the best ones at evaluation time. We follow single-path uniform sampling strategy (Guo et al., 2020), by sampling a pooling configuration $c \in C$ (from the search space described above) with uniform probability at each iteration. For a mini-batch of training samples and the corresponding annotations $(x, y) \in \mathcal{X}_{tr}$, we update the network weights $w$ by minimizing the following loss:

$$\sum_{(x,y) \sim \mathcal{X}_{tr}, c \sim C} \mathcal{L}(f_c(x, w), y), \tag{1}$$

where $f_c$ is the output of the network for a given pooling configuration $c$ and $\mathcal{L}$ is a classification loss such as cross-entropy. The choice of uniform sampling (Xie et al., 2018; Guo et al., 2020) is hyperparameter free and ensures fairness in training among architectures. As the configurations are chosen uniformly, this training does not favor any specific configuration provides a meaningful estimation of the performance of each configuration.

At the end of training, the network $f$ is evaluated on a validation set $\mathcal{X}_{val}$ for all configurations $C$, and top-k configurations with higher accuracy are selected as best configurations. Previous work has shown that this approach works when used to select network parts that do not share weights, however, in our setting, as all configurations share the same parameters, they produce interference,

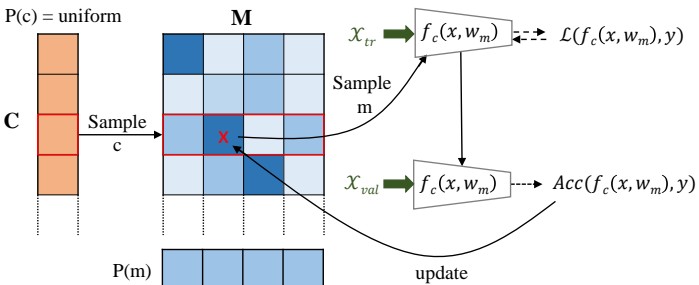

Figure 2: Balanced Mixture of SuperNets. At each training iteration we uniformly sample a pooling configuration $c$, then a model with a probability proportional to $p(m|c)$. The model weights $w_m$ are updated on a mini-batch of training data from model accuracy $Acc$ on validation data. A moving average of the accuracy is used to update $p(c, m)$ such that $p(m)$ remains a uniform, balanced mixture of models, ensuring that each model is trained for the same number of iterations.

and the SuperNet is no longer a good proxy to find the best performing configurations, which is the aim of this approach. This is illustrated in Fig. 1, where jointly training two pooling configurations produces worse results than training either one or the other independently. In fact, the features and structures seen by the convolutional filters when working with different pooling configuration are drastically different and learning them together hinder performance. For this reason, to reduce the weight sharing and to avoid the interference of different configurations on the same model we propose to use a mixture of models.

## 2.3 Balanced Mixture of SuperNets

Instead of using a single set of weights or SuperNet, we propose to use $M$ independent SuperNet models or weight sets $w_m$ associated with a network $f_c$. In this way, each set of weights may specialize to represent unique subsets of specific pooling configurations, leading to improved performance. After each mini-batch training iteration, we compute the moving average of the accuracy $a_{c,m}$ on a validation minibatch $\mathcal{X}_{val}$ for a given network $f_c(\cdot, w_m)$ with pooling configuration $c$ and weight set $w_m$ as follows:

$$a_{c,m} = \beta\, a_{c,m} + (1 - \beta)\, Acc(f_c(x, w_m), y), \quad (x, y) \sim \mathcal{X}_{val}, \;\; c \sim \mathcal{C}, \;\; m \sim p(m|c) \qquad (2)$$

where $\beta$ is a hyper-parameter controlling the smoothness of the moving average. At each iteration, the pooling configuration $c$ is sampled uniformly, while the model $m$ is sampled based on the conditional probabilities $p(m|c) = \frac{p(c,m)}{\sum_c p(c,m)}$. The probability $p(c, m)$ is computed by normalizing the accuracies $a_{c,m}$ with a $\tau$-softmax function:

$$p(c, m) = \frac{\exp(a_{c,m}/\tau)}{\sum_{j,k} \exp(a_{j,k}/\tau)}, \qquad (3)$$

where in Equation (3) $\tau$ is a temperature hyperparameter of the probability distribution where $\tau \to 0$ implies a maximally concentrated distribution. These probabilities are thus proportional to the accuracy of the chosen joint configuration of pooling $c$ and model $m$. We could use directly these probabilities to sample with a multinomial distribution a joint configuration $(c, m)$ to train a mini-batch. However, this would make the model focus on some specific joint configuration/model during training and will lead to coupling of pooling configurations and models due to unbalanced sampling. Instead we want the training to give equal importance to each pooling configuration $c$ while selecting the most promising model. The best pooling strategy is then selected at the end of the training, making sure that each configuration and each model have received the equal amounts of training.

We thus achieve balance SuperNet mixtures by imposing the constraint that the joint probability distribution $p(c, m)$ have uniform marginals, i.e. $\sum_i p(c_i) = 1/C$ and $\sum_j p(m_j) = 1/M$. We use the iterative proportional fitting (IPF) algorithm to achieve this, where $p(c, m)$ is alternatingly normalized along $c$ and $m$ dimensions until uniformity is achieved. The KL-distance is used to estimate the deviation of $p(m)$ from uniformity, and IPF terminates when the KL-distance falls below the threshold of $\delta = 0.0001$. At this point the pooling configuration $c$ is sampled uniformly while the model $m$ is sampled from the conditional distribution $p(m|c)$.

Balancing allows each model to focus on different configurations, while ensuring equal importance of all models during training iterations. $\tau$ is a concentration parameter and is decreased linearly over the course of training. After training, the mixture of SuperNets is used to select the top-k performing configurations, by evaluating the model $m$ with highest $p(m|c)$ for each configuration $c$ on the validation data. In this way, the number of evaluations required depends only on the number of configurations even if many models are considered. The number of configurations to evaluate may still become prohibitive when using very deep models (such as ResNet50). In this case, instead of evaluating all configurations, we can use $p(c, m)$ as a proxy to select the correct model and $a_{c,m}$ to rank configurations and evaluate only the top ranking on the entire validation set, and therefore reducing the computation required to select the best model after training.

## 3 Experiments

In this section we perform several experiments and ablations in order to evaluate the performance of our proposed approach. We first individually train and evaluate the performance of a small ResNet with 36 different pooling configurations on CIFAR10 and show that the optimal pooling can improve the performance of the model. We also compare the correlation between different configurations of a Mixture of SuperNets for various number of mixtures (M) with the individually trained configurations and demonstrate that more models help in obtaining a better correlation. Next, we present an ablation, considering different variants of weight sharing for DARTS and Single Path One-Shot (SPOS) approaches. Additionally, we compare our model with other NAS and non-NAS approaches. Finally, we evaluate our model on a higher resolution dataset (Food101), with a larger model (ResNet50). In all experiments, we separate the default training set of each dataset in 50% for training and 50% for validation, used for estimating the quality of the configurations.

### 3.1 Performance of individually trained Models

As shown in (Riad et al., 2022) the pooling configuration of a CNN has a large impact on the performance of the model. To establish a benchmark, we consider all possible pooling configurations that satisfy conditions set in section 2.1 to avoid useless configurations. With 2 pooling operations and 9 available pooling locations, the search space size is combination $\binom{9}{2} = 36$. This limited search space facilitates the exhaustive search of entire space and allows us to find the true ranking by training independently all baseline models. In other words, unlike (Su et al., 2021a; Chau et al., 2022) benchmarks, we fully train all architectures.

For this evaluation, we choose a lightweight ResNet configuration (He et al., 2016) and exhaustively train each architecture 3 times with different seeds and report the average result. The complete results on the entire space are included in tab. 6 in appendix. The standard pooling configuration is configuration [4,3,3], which has the first pooling layer after 4 ResNet block and the second after other 3 blocks and its classification accuracy is 90.52% ± 0.6. In contrast, the best configuration is [6,1,2], with an accuracy of 92.01% ± 0.12. This shows that even for one of the most common datasets, the pooling structure is not optimal, and therefore it makes sense to propose models that can optimize the CNN pooling configuration. We hope that this benchmark will motivate researcher in the field to not overlook the importance of an optimal pooling configuration. While this benchmark is relatively small, with only 36 feasible pooling configurations, we show in

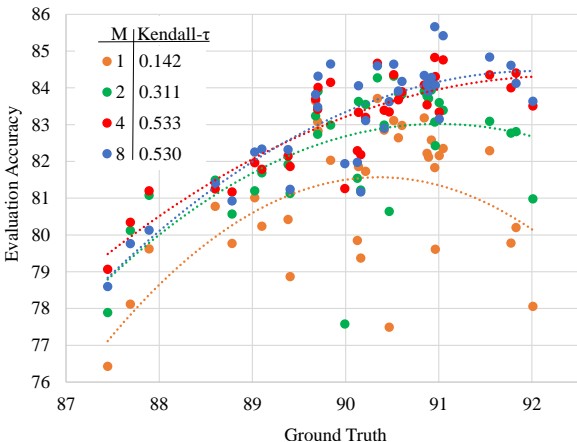

Figure 3: Evaluation accuracy vs. ground truth accuracy for CIFAR10 test dataset for different number of mixtures (M). Each point represents the performance of a given configuration with a model trained independently (Ground truth - x axis) and the same configuration evaluated with the SuperNet (Evaluation Accuracy - y axis) with different numbers of mixtures M (colors). Rank correlation measured by Kendall's tau increases with number of models.

the next experiments that it is quite challenging and most of the commonly used NAS methods fail to find a good pooling configuration.

## 3.2 Balanced Mixture of SuperNets

We evaluate our proposed balanced mixture of SuperNets on our benchmark with different numbers of models M=[1,2,4,8]. The case of M=1 is equivalent to SPOS method with uniform sampling as in (Guo et al., 2020) with complete weight sharing among architectures. In Fig.3, we show the correlation of the performance obtained by our SuperNet trained with different number of mixture models, with true accuracies of given pooling configuration models trained independently on test set, and we calculate Kendall tau-rank correlation coefficient. As expected, using multiple mixtures shows overall stronger correlation compared to SPOS uniform sampling ($M = 1$). This model has poor correlation with ground-truth and is unable to find optimal configurations even in a limited search space. The improvement is more prominent with higher ranking models, resulting in finding better final configurations. For this experiment, for more than M=4 mixture models the gain in performance seems to saturate.

## 3.3 Relaxing the full weight sharing

We argued that one reason that makes the learning of optimal pooling difficult is the fact that the model weights are fully shared, i.e. the same weights are used for all feature scales/resolutions. In this subsection, we consider the case of relaxing the weight sharing and using a different weights for each resolution. For this experiment we evaluate SPOS and DARTS, in case of using the same parameters for each feature map resolution (*Fully shared*) or different (*Not shared*). For SPOS, we consider two different variants. The first uses the 36 paths that are meaningful. However, we note that by using only those 36 paths and different filters per resolutions would induce some filters to be trained much more than others, which would bias the selection of the optimal filters. To avoid that, we also considered a case in which all 19,683 possible configurations are used. In this case the training takes longer and has more noise. For DARTS, in the case in which each resolution has different parameters (*Not shared*) we consider two variants, the case of initializing filters with the same initialization for all resolutions (*same init.*) or different (*rnd init.*). As tab. 1 shows, only our adaptive association of pooling configurations and model parameters (*Balanced Mixtures*) manage to obtain better results than the *Default* pooling configuration.

Table 1: CIFAR10 results for different search methods, number of model weights and paths. For DARTS we consider a model with shared weights for different feature map resolutions (*Fully Shared*) and not shared (*Not Shared*) with different weights per resolution with either same or different initialization. In all cases accuracy is lower than the *Default*. For SPOS, we test *Fully Shared* weights and *Not Shared* with different number of paths. Results are comparable to the default setting. Only our *Balanded Mixtures* of SuperNets clearly outperforms default.

| NAS Method | Mixtures(M) | Paths | Architecture | Accuracy |
|---|---|---|---|---|
| *Default* | 1 | 1 | [4,3,3] | 90.52 ± 0.1 |
| DARTS | | | | |
| *Fully shared* | 1 | 19,638 | Fig. 4a | 89.23 ± 0.13 |
| *Not shared - same init.* | 4 | 19,638 | Fig. 4b | 89.85 ± 0.18 |
| *Not shared - rnd. init.* | 4 | 19,638 | Fig. 4b | 90.03 ± 0.21 |
| SPOS | | | | |
| *Fully shared* | 1 | 36 | [3,3,4] | 90.61 ± 0.17 |
| *Not shared* | 4 | 36 | [4,2,4] | 90.34 ± 0.12 |
| *Not shared* | 4 | 19,683 | [4,2,4] | 90.34 ± 0.12 |
| *Balanced Mixtures (Ours)* | 4 | 36 | [5,3,2] | **91.55 ± 0.08** |

Table 2: CIFAR10 found architectures, accuracies and training time for different search methods. Results on DARTS are relaxed selections of resolutions and therefore outside the search space we defined in our work.

| NAS Method | Architecture | Accuracy | Training (GPU hours) |
|---|---|---|---|
| DARTS + GAEA | Fig. 4a | 89.12 ± 0.1 | 12 |
| DARTS | Fig. 4a | 89.23 ± 0.08 | 12 |
| SBE + Unif. Smp. | [1,6,3] | 90.13 ± 0.06 | 2.5 |
| SPOS | [4,2,4] | 90.34 ± 0.12 | 1.5 |
| MCTS UCB | [4,2,4] | 90.34 ± 0.12 | 2.5 |
| SBE | [2,3,5] | 90.42 ± 0.08 | 2 |
| Default | [4,3,3] | 90.52 ± 0.10 | - |
| MCTS UCB + Unif. Smp. | [4,4,2] | 90.85 ± 0.09 | 2.5 |
| Balanced Mixtures (Ours) | [5,3,2] | **91.55 ± 0.08** | 6 |
| Best conf. (Bruteforce) | [7,1,2] | 92.01 ± 0.12 | 98 |

## 3.4 Comparison with NAS-based methods

We compare our method with several variants of commonly used NAS methods: Differentiable architecure search (DARTS) (Liu et al., 2018b), GAEA (Li et al., 2020b), Monte-Carlo Tree Search (MCTS) (Su et al., 2021a) and Boltzmann Softmax Exploration (BSE) (Asadi and Littman, 2017; Cesa-Bianchi et al., 2017). An explanation of these approaches is presented in Appendix C.

In tab.2, we present results in terms of found architecture and accuracy of the selected configuration. Even if the number of possible configurations is limited, none of the method manages to obtain the best pooling configuration, which is 1.5 points above the default baseline. DARTS-based methods (as they do not have constraints on the pooling configurations) yield strange configurations in which down-sampling if followed by up-sampling (see Fig.4a) which brings a loss of information and therefore poor results. Other methods based on SPOS, BSE and MCTS with different variants, obtain results that are comparable and close to default setting. Our method with M=4 models is the only one that approaches the optimal performance, with an accuracy of 91.55%.

## 3.5 Comparison with other methods

We compare our Balanced Mixture of SuperNets with other approaches that aim to improve performance by learning the scale of the feature representation through different algorithms not

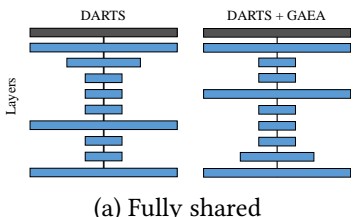
(a) Fully shared

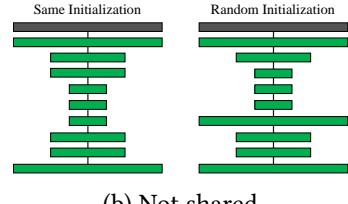
(b) Not shared

Figure 4: Final configurations found by DARTS Liu et al. (2018b) and its variant (Li et al., 2020b). The first layer in grey is fixed at the maximum input resolution. (a) shared weights per layers. (b) different weights per feature map resolution for each layer, weight are initialized randomly or with same values.

Table 3: Accuracy comparison between default, different methods that find optimal feature map scale and our method on CIFAR10 and CIFAR100 for ResNet18 and ResNet50.

| Method | Dataset | Backbone | Baseline | Improved | Gap |
|---|---|---|---|---|---|
| DiffStride (Riad et al., 2022) | CIFAR10 | ResNet18 | $91.4 \pm 0.2$ | $92.4 \pm 0.1$ | 1.0 |
| Balanced Mixtures (Ours) | CIFAR10 | ResNet18 | $90.45 \pm 0.21$ | $91.51 \pm 0.09$ | 1.06 |
| DynOPool(Jang et al., 2022) | CIFAR100 | ResNet50 | 78.50 | 80.60 | 2.1 |
| ShapeAdaptor(Liu et al., 2020) | CIFAR100 | ResNet50 | 78.50 | 80.29 | 1.8 |
| Balanced Mixtures (Ours) | CIFAR100 | ResNet50 | $77.57 \pm 0.18$ | $79.61 \pm 0.21$ | 2.04 |

based on NAS. In contrast to the other experiments, here we present results provided directly by other papers. In this case, we noticed that the final performance is highly affected by the performance of the baseline model, which can vary depending on small and difficult to control details. Thus, in order to make the comparison fairer, results of the method (*Improved*) are presented with respect to the corresponding *Baseline*, so that we can consider not only the absolute performance but also the relative *Gap* with respect to the baseline. In tab. 3 we compare our results with DiffStride (Riad et al., 2022) on CIFAR10 with ResNet18 and CIFAR100 with ResNet50. We also compare with DynOPool (Jang et al., 2022) and Shape Adaptor (Liu et al., 2020) on CIFAR100 with ResNet50. In all experiment our approach performs comparable to other methods that explicitly change and improve the pooling layers.

## 3.6 Larger Dataset and Model

To evaluate our method on new domains, we use fine-grained food classification on Food101, which contains more images than CIFAR and at higher resolution. We adapt a deeper ResNet network, ResNet50 and fix first layer and initial downsampling in the architecture. By using a deeper network, the search space size is increased to combination $\binom{15}{3} = 455$ architectures. We conduct our experiments on input image resolution of 256. Food101 is a challenging fine-grained object classification dataset that consists of 101 food categories with 75,750/25,250

Table 4: Resnet50 on Food101. We report best architectures, their accuracy after retraining for different number of Mixtures (M) of our SuperNets. Increasing M leads to an architecture with better accuracy.

| Models | Best Arch. | Accuracy |
|---|---|---|
| Default | [3,4,6,3] | $84.00 \pm 0.10$ |
| M = 1 | [6,4,5,1] | $84.24 \pm 0.09$ |
| M = 2 | [4,5,6,1] | $84.34 \pm 0.18$ |
| M = 4 | [8,3,3,2] | $84.35 \pm 0.14$ |
| M = 8 | [9,4,2,1] | $84.73 \pm 0.09$ |

training/test split. We show the results in tab. 4 The results clearly show increased improvement by increasing number of models. Food101 has high intra-class variance, that does not show distinguishing spatial layout and the classification would need to rely on colors, textures and local information to distinguish them (Bossard et al., 2014). Therefore unsurprisingly, identified architectures show a tendency towards adopting high resolution feature maps in early layers.

## 4  Conclusion

In this paper we presented the problem of learning the optimal scale for CNN feature maps by learning pooling/stride configurations. We showed that current NAS methods (single-path uniform sampling, differentiable methods and tree search) are insufficient for this problem. We have established empirically the importance of appropriate search space design by an extensive evaluation on CIFAR10 and introduced a balanced mixture of SuperNets to alleviate the weight-sharing poor ranking correlations for this problem. Finally, we compared our method with several non NAS-based approaches and evaluated it on a more challenging dataset and larger model and search space.

## 5  Broader Impact

Our approach requires a higher number of iterations needed to converge, due to joint use of multiple models. However, the computational cost and memory requirements of one iteration are not affected as for each minibatch we select only one pooling configuration and one model. Also, we should consider that even though the balanced mixtures approach can potentially work for any NAS problem, we evaluated it only for finding the optimal pooling configuration, which was the focus of this work. We leave a more general evaluation and application of the approach to future work. One could argue that NAS in general are a waste of computation, however, they help to avoid an even more expensive validation search for the optimal hyper-parameters of the model.

### Acknowledgments

This research was supported in part by " Natural Sciences and Engineering Research Council of Canada" (NSERC) and "Digital Research Alliance of Canada" (alliancecan.ca)

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

## A  Related Work

**Optimal Scale for CNN Feature Maps** Some recent work have addressed the optimization of feature map scales in CNN design by adapting dynamic kernel shapes (Romero et al., 2021; Pintea et al., 2021), learning resizing modules (Liu et al., 2020; Riad et al., 2022; Jang et al., 2022) or receptive field analysis (Richter and Pal, 2022). Feature map sizes as well as receptive field are controlled by kernel sizes, and number and position of down-sampling layers in a CNN.

Among the works that indirectly learn feature map scales by learning kernel sizes, N-Jet (Pintea et al., 2021) uses Gaussian derivative filters to dynamically adapt kernel size during training, using scale space theory and employs a safe-sub-sampling alternative. Flexconv (Romero et al., 2021) learns long range dependencies without using pooling and replaces them with multiplication of continuous kernels and a Gaussian mask and learns the parameters of the mask. However, these methods often struggle with large kernel sizes(Pintea et al., 2021) or use uses several techniques such as Fourier transformation and training on down-sampled images to reduce the cost (Romero et al., 2021).

Another work introduces a resizing module, ShapeAdaptor (Liu et al., 2020) that learns the scale and mixing weight of linear combination of two feature map sizes in differentiable manner. However, the number of shape adaptor modules is a fixed hyperparameter that limits the placement of down-sampling layers, and the framework only works with max-pooling operations. DiffStride (Riad et al., 2022) proposes a down-sampling layer with learnable fractional strides by casting down-sampling in spatial domain as cropping in frequency domain and show that on CIFAR10 and CIFAR100 dataset with ResNet18, lower layers tend to preserve more details while the pooling is performed more aggressively at later layers. However, the placement of down-sampling layers remains fixed and the pooling is performed in spectral domain resulting in higher cost and involving complex valued operations that are not optimized on GPU. DynOPool Jang et al. (2022) relaxes the problem by using bi-linear interpolation to allow for non integer feature map sizes and learns the resizing scale in differentiable manner. In these models, learning the resizing factor and network weights are performed simultaneously on training set, which can render the optimization hard and introduce bias towards non-optimal solutions (He et al., 2021). Furthermore, these approaches are outside the NAS framework often cannot easily find a single architecture and might introduce additional costs at run-time.

**Macro Search in NAS** While early NAS works focused on a global search space Zoph and Le (2016), containing both micro and macro search space, the complexity of the search space contributed to their great computational cost. NASNet (Zoph et al., 2018) proposed a cell-based search space, reducing the search space size from the entire network to only a set of operations in cells with many recent works (Liu et al., 2018b,a; Real et al., 2019; Zhong et al., 2018; Ying et al., 2019; Dong and Yang, 2020; Siems et al., 2020) using cell-based search spaces due to its efficiency.

In terms of search strategy (Zoph and Le, 2016; Pham et al., 2018; Baker et al., 2016; Cai et al., 2018) use reinforcement learning, while (Elsken et al., 2018; Lu et al., 2019; Real et al., 2017; Lopes and Alexandre, 2022; Xie and Yuille, 2017) use evolutionary algorithms (EA) and (Hu et al., 2019, 2018) use random search to perform macro search. Several of the early works (Baker et al., 2016; Zoph and Le, 2016; Real et al., 2017) however, required hundreds of GPUdays to perform the search. To navigate the huge search space, progressive NAS methods (Liu et al., 2018a; Chen et al., 2019) were proposed that progressively add layers to the a shallow network, while (You et al., 2020; Li et al., 2020a) drop unpromising architectures progressively. Among works that perform NAS on the entire network, MCTS (Su et al., 2021a) uses Monte-Carlo tree search algorithm to establish a benchmark in MobileNetV2 search space. While, this model is not cell-base, the feature map sizes and downasmpling locations are still fixed. Some of these works (Su et al., 2021a; Xia et al., 2022) search on all layers of a CNN, however they still use a fixed template for the CNN's outer-skeleton.

DenseNAS (Fang et al., 2020) proposes a densely connected search space by designing routing blocks. The routing blocks contain shape alignment layers that perform convolutions on different shaped inputs (channels and spatial dimension) and are the sum of the results. Both, basic blocks (containing operations) and routing blocks are relaxed and perform a gradient based optimization of mixing weights. Several blocks at the same resolution are searched as well as number of channels per layer. The position to downsampling is determined along with the block count search.

TNAS (Qian et al., 2022) factorizes the space in a hierarchical manner by designing an operation space (by a binary operation tree), and architecture layers (by a architecture tree) and performs a bilevel search on both trees. LCMNAS (Lopes and Alexandre, 2022) autonomously generates search spaces by creating weighted directed graphs with hidden properties from existing architectures and performs search using EA and evaluation using a performance predictor. Compared to these approaches, ours is more general, the models are specialized automatitically and does not require prior knowledge about search space.

**Weight Sharing in NAS** The problem of how to efficiently evaluating candidate architectures has been a bottleneck of NAS research (Ren et al., 2021; Cha et al., 2022). Training each candidate architecture from scratch to convergence provides the true performance of the architecture, however that was one of the reasons for significant cost of early NAS methods (Baker et al., 2016; Zoph and Le, 2016; Real et al., 2017; Xie and Yuille, 2017). A great improvement in this regard was using weight sharing (Pham et al., 2018) among architectures in one-shot methods, where the search space is defined as an oveparametrized SuperNet, from which every possible architecture can be derived. After training the SuperNet, the candidate architectures are evaluated without any additional training by inheriting the weights from the SuperNet. One of the most influential works that is based on SuperNet training is DARTS (Liu et al., 2018b), which relaxed the discrete search space of NAS, and enabled using backpropagation to jointly learn SuperNet weights and architecture parameters. However, several works show that the architecture parameters fail to reflect the importance of them (Wang et al., 2021; Yu et al., 2019; Zhou et al., 2021) as well as facing challenges in generalizing (Chen et al., 2019; Li et al., 2020a; Xie et al., 2018; Yu et al., 2019) and stability (Chen and Hsieh, 2020; Wang et al., 2021; Arber Zela et al., 2020; Zhang et al., 2021) and high memory requirements to perform backward pass through all configurations. Poor rank correlation is the result of coupling between the architecture and network weights as well as coupling of weights among architectures. Training architecture weights simultaneously results in introducing bias by favoring certain architectures during training. These weights can be decoupled by uniform sampling and single path methods (Guo et al., 2020), which has been shown to outperform training a SuperNet as a whole as in DARTS. One direction to reduce one-shot methods suffer from poor rank correlation (Yu et al., 2019) and performance degradation due to the co-adaptation of weights among architectures (Bender et al., 2018), is reducing the amount of weight shared among architectures.

Among works that directly reduce weight sharing, few-shot NAS (Zhao et al., 2021; Su et al., 2021b) was proposed to partition SuperNet to multiple sub-SuperNets. The split is performed by random selection of an edge in SuperNet and dividing all operation on that edged into sub-SuperNets. While this setup reduces weight sharing and improves the performance over one-shot methods, the partitioning criteria is inefficient as it fails to identify similar and dissimilar models and whether the partitioning of specific regions results in any meaningful gain. To address this issue, (Hu et al., 2022) address these issues by proposing a gradient matching score that decides which candidate network should share weights, while (Liu et al., 2022) propose a gradual training from one-shot to few-shot NAS. However, these works focus on finding operations in a micro-search space, the SuperNet partitioning is not automatic (Zhao et al., 2021), and focus on NAS benchmarks that are not applicable to the specific application addressed in this work.

Table 5: Search space design for experiments conducted in paper. We consider the same number of downsampling operations as a default network (no. pooling) and exclude from the search space the first pooling layer as the it corresponds to a manipulation of the input image. For larger input images (ImageNet and Food101), we keep the first layer (conv and maxpooling) predefined for computational efficiency and only search the pooling locations among layers after the predefined maxpooling layers. The search space size is then the combination $\binom{Layers-1}{pooling}$

| Backbone | Dataset | Searched feature map sizes | no. searched layers | no. pooling | Search Space Size |
|---|---|---|---|---|---|
| ResNet20 | CIFAR10 | [32,16,8] | 10 | 2 | 36 |
| ResNet18 | CIFAR10 | [32,16,8,4] | 9 | 3 | 56 |
| ResNet50 | CIFAR100 | [32,16,8,4] | 17 | 3 | 560 |
| ResNet50 | Food101 | [56,28,14,7] | 16 | 3 | 455 |
| ResNet18 | ImageNet | [56,28,14,7] | 8 | 3 | 35 |

## B Experimental Setup and Details

### B.1 Datasets and Hyperparameters

All datasets in our experiments were split 50/50 for NAS training and validation. Unless otherwise specified, all experiments were run 3 times with random seeds and average and standard deviations are reported. For ResNet18 and Resnet50 tests we use mixed-precision operations and FFCV (Leclerc et al., 2022) library to increase training efficiency.

We tuned the hyperparameters either by grid search for our experiments or when compared with other work, used similar hyperparameters. We used SGD with learning rate scheduling and weight decay for all our experiments. For ResNet20 we used learing rate of 0.1 with cosine annealing and weight decay 1e-3 and batch size 256. For DARTS experiments, we used Adam for architecture parameters with learning rate 1e-2. For our Balance Mixture of SuperNets, the number of training epochs is set proportional to number of models to ensure sufficient training. Furthermore, we initialize $\tau = 1$ and decrease it linearly during the training with minimum value of $1/(100M)$ with $M$ the number of models. For experiments on CIFAR10 and CIFAR100 with ResNet18, we train for 400 epochs, with learning rate of 0.1 and reduced it by factor of 0.1 on epochs [200,300] and weight decay of 5e-3. For ResNet50 experiments on CIFAR100 we trained for 250 epochs and changed the scheduling to reducing by factor 0.2 at epochs [60,120,160]. For Food101 we used learning rate of 0.1, cosine annealing and batch size 256.

### B.2 Search Space Details

Summary of search space design for the experimetns is provided in tab. 5. ResNet20 (He et al., 2016) architecture consists of one convolutional layer followed by 9 ResNet layers. The original structure consists of [32, 16, 8] feature map sizes and [16, 32, 64] number of filters respectively, with [3,3,3] blocks per resolution. In our implementation, fully connected layer is removed and strided convolutions are replaced by maxpooling.

## C Comparison With Other NAS Methods (Details)

**DARTS:** Differentiable approaches first proposed by DARTS (Liu et al., 2018b) has been commonly used in recent years for NAS problems. As one of the most efficient and reliable NAS methods, we utilize DARTS for our problem. In terms of optimization, we use the same differentiable principle for architecture search as DARTS (Liu et al., 2018b). Instead of learning weights for different networks branches, we learn weights for different feature map resolutions by learning associated architecture parameter $\alpha$. Since changing the position of the pooling layer in the network changes the size of a feature map and therefore it invalidates all the subsequent blocks, we need to find a way to achieve this without without changing the feature map resolution. Therefore, we introduce

a multi-resolution block $M$ defined as the weighted combination of convolutions $V$ at different resolutions $r$ of the same filters $f_l$ at a given layer $l$:

$$h_{l+1}(x, y) = \mathbf{M}(h_l(x, y))$$
$$= \sum_{r=1}^{R} \alpha_{l,r} \mathbf{U}_{2^r}(\mathbf{V}(\mathbf{S}_{2^r}(h_l(x, y)), f_l(h, w))). \tag{4}$$

The resulting feature map is the sum of the feature map at each resolution multiplied by a coefficient $\alpha_{l,r}$ and rescaled to the initial feature map resolution $(x, y)$ resolution with an upsampling operation $\mathbf{U}$.

The normalized coefficient $\alpha_{l,r}$ learns the relative strength of a certain resolution with respect to the others for a given layer $l$ and is computed as a softmax over feature map resolutions:

$$\alpha_{l,r} = \frac{\exp(\alpha_{l,r})}{\sum_s \exp(\alpha_{l,s})}. \tag{5}$$

This approach allows us to train a model that can learn the convolutional filters, but at the same time, with a marginal increase in computation and memory can also learn the best feature resolution to use at each layer.

To make the search space as similar to out method as possible we manually select highest resolution for first layer of the network, however imposing further restrictions on the search space is more difficult. At the end of training we select maximum $\alpha_{l,r}$ for each layer as final architecture to retrain. We used ADAM (Kingma and Ba, 2014) optimizer to train $\alpha$ and SGD with cosine learning rate for CNN weights. Furthermore, We used another standard optimizer and GAEA(Li et al., 2020b) which both fail on this task as it finds sub-optimal architectures. As seen in section 3.5, the architectures found by this approach are atypical for classification task as they utilize upsampling in several later layers, resulting in lower accuracy of final architecture.

**Monte-Carlo Tree Search**: Several recent works (Su et al., 2021a; Wang et al., 2020) have used MCTS for NAS problem both in training and search stage of NAS. As MCT captures dependencies amongst layers (Su et al., 2021a), it is viable candidate for our task. We designed the same search space as 2.1 as a binary tree. Each layer $l$ of CNN correspond to generations of tree, at each layer maximum of two nodes exist, 1) same resolution and 2) downampling . By fixing the leaf nodes at minimum resolution and first layer at highest resolution, we design an asymmetric tree with 36 leaves.

To balance exploration and exploitation we use Upper Confidence Bound (UCB) to calculate sampling probabilities as:

$$UCB(r_i^l) = \frac{a(r_i^l)}{n_i^l} + c\sqrt{\frac{log(n_p^{l-1})}{n_i^l}} \tag{6}$$

Where $p$ corresponds to the parent node and $a$ is the reward and $c$ is a hyperparameter constant controlling the trade-off. In our experiments we use validation accuracy on minibatches as reward.

We considered two settings: sampling with UCB from the beginning and sampling with a uniform warm-up. By using UCB it is expected that training will focus more on better performing architectures compared to uniform sampling, therefore improving the ranking of top architectures. At the end of the training phase we evaluate the found architecture by on validation set. Results in tab. 2 shows MCT finds sub-optimal architecture in both cases.

**Boltzmann Softmax Exploration (BSE)**: BSE is one of the simplest reinforcement learning exploration strategies. For sampling an architecture $c$ we use:

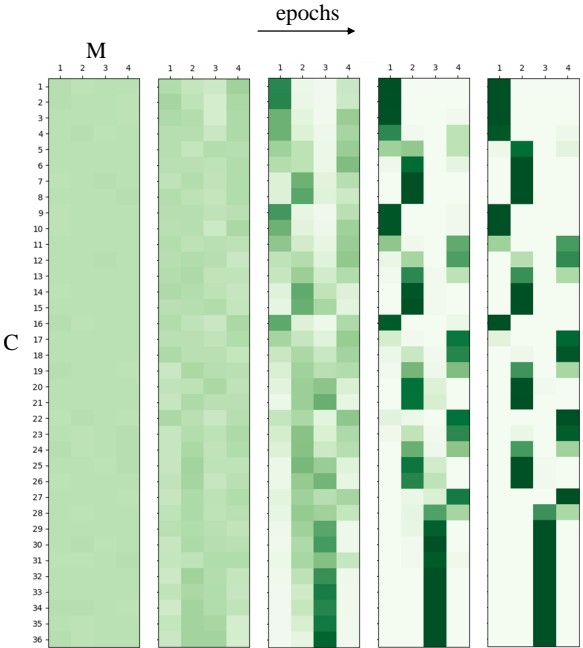

Figure 5: Progress of $p(c|m)$ during training for a mixture of M=4 SuperNets and 36 pooling configurations. As temperature $\tau$ in equation 3 is linearly decreased, the distribution transitions from uniform (left) to concentrated (right).

$$p(c) = Softmax(\tau\, a_c) \qquad (7)$$

Where $p(c)$ is the probability of selecting architecture $c$, and $a_c$ is the reward (here the validation accuracy) and $\tau$ is the inverse temperature, controlling exploration and exploitation. We increase $\tau$ from 1 linearly during the experiment.

With the defined search space of 36 architectures, we choose a linear schedule for the inverse temperature that balances exploration and exploitation. Furthermore we considered another option with a uniform warm-up. However, the appropriate scheduling is difficult as the model can either continue to explore sup-optimal solutions or commit to one solution too early (Cesa-Bianchi et al., 2017).

## D  Comparison With Other Methods (Details)

### D.1  Comparison With DiffStride, DynOPool and ShapeAdaptor

To compare with DiffStride (Riad et al., 2022) we ResNet18 architecture where original structure consists of 8 blocks With 4 resolutions as [2,2,2,2], resulting in the search space of 56 configurations.

To compare with DynOPool (Jang et al., 2022) and Shape Adaptor (Liu et al., 2020) we used ResNet50. Since ResNet50 for CIFAR does not include initial downsampling layers, the search space consist of 560 configurations with default configuration of [4,4,6,3]. It should be noted that the search space of these methods are not identical with ours.

## E  Extended Results

In tab. 6 we present ground truth accuracy of all configurations in our searrch space for ResNet20, as described in 3.1. In figure 5 we show the progress of $p(m|c)$ for our method described in 2.3.

Table 6: CIFAR10 accuracies for all configurations with ResNet20 Backbone. Architectures are displayed in terms of number of blocks associates with feature map sizes of [32, 16, 8]. Architecture 24 is the original ResNet20 architecture pooling configuration.

| no. | Architecture | Accuracy | no. | Architecture | Accuracy |
|-----|-------------|----------|-----|-------------|----------|
| 1 | [ 1 , 1 , 8 ] | 87.45 ± 0.06 | 19 | [ 3 , 4 , 3 ] | 90.92 ± 0.1 |
| 2 | [ 1 , 2 , 7 ] | 87.69 ± 0.08 | 20 | [ 3 , 5 , 2 ] | 90.88 ± 0.08 |
| 3 | [ 1 , 3 , 6 ] | 87.89 ± 0.17 | 21 | [ 3 , 6 , 1 ] | 90.14 ± 0.16 |
| 4 | [ 1 , 4 , 5 ] | 88.6 ± 0.15 | 22 | [ 4 , 1 , 5 ] | 89.68 ± 0.14 |
| 5 | [ 1 , 5 , 4 ] | 89.38 ± 0.07 | 23 | [ 4 , 2 , 4 ] | 90.34 ± 0.13 |
| 6 | [ 1 , 6 , 3 ] | 90.13 ± 0.14 | 24 | [ 4 , 3 , 3 ] | 90.52 ± 0.15 |
| 7 | [ 1 , 7 , 2 ] | 90.16 ± 0.1 | 25 | [ 4 , 4 , 2 ] | 90.85 ± 0.12 |
| 8 | [ 1 , 8 , 1 ] | 89.41 ± 0.15 | 26 | [ 4 , 5 , 1 ] | 89.71 ± 0.16 |
| 9 | [ 2 , 1 , 7 ] | 88.78 ± 0.1 | 27 | [ 5 , 1 , 4 ] | 91.05 ± 0.15 |
| 10 | [ 2 , 2 , 6 ] | 89.03 ± 0.12 | 28 | [ 5 , 2 , 3 ] | 90.96 ± 0.15 |
| 11 | [ 2 , 3 , 5 ] | 90.42 ± 0.1 | 29 | [ 5 , 3 , 2 ] | 91.55 ± 0.09 |
| 12 | [ 2 , 4 , 4 ] | 90.57 ± 0.11 | 30 | [ 5 , 4 , 1 ] | 89.84 ± 0.08 |
| 13 | [ 2 , 5 , 3 ] | 90.89 ± 0.07 | 31 | [ 6 , 1 , 3 ] | 91.78 ± 0.11 |
| 14 | [ 2 , 6 , 2 ] | 91.01 ± 0.13 | 32 | [ 6 , 2 , 2 ] | 91.83 ± 0.13 |
| 15 | [ 2 , 7 , 1 ] | 90.22 ± 0.18 | 33 | [ 6 , 3 , 1 ] | 90.96 ± 0.12 |
| 16 | [ 3 , 1 , 6 ] | 89.1 ± 0.06 | 34 | [ 7 , 1 , 2 ] | 92.01 ± 0.18 |
| 17 | [ 3 , 2 , 5 ] | 89.7 ± 0.1 | 35 | [ 7 , 2 , 1 ] | 90.47 ± 0.11 |
| 18 | [ 3 , 3 , 4 ] | 90.61 ± 0.17 | 36 | [ 8 , 1 , 1 ] | 89.99 ± 0.12 |

Table 7: Resnet18 on ImageNet. We report best architectures, their accuracy after retraining for different number of Mixtures (M) of our SuperNets. As the ranking is noisy, we retrained the best 3 architectures based on our ranking and report top-1 and top-3 accuracies.

| Models | top-1 Arch. | top-1 | top-3 Best | top-3 average |
|--------|-------------|-------|------------|---------------|
| Default | [2,2,2,2] | 68.32 ± 0.24 | NA | NA |
| M = 1 | [1,3,1,3] | 62.21 ± 0.26 | 65.91 ± 0.21 | 63.51 ± 1.56 |
| M = 2 | [3,1,1,3] | 62.56 ± 0.18 | 68.32 ± 0.24 | 64.70 ± 2.57 |
| M = 4 | [5,1,1,1] | 65.88 ± 0.24 | 66.12 ± 0.18 | 64.73 ± 1.78 |
| M = 8 | [2,3,2,1] | 64.81 ± 0.11 | 66.12 ± 0.23 | 64.15 ± 2.98 |

## E.1 Experiment on ImageNet (Deng et al., 2009)

We used ResNet18 architecture as backbone for ImageNet dataset. We trained the top-1 and top-3 best architectures found by our method with (N=1,2,4,8) for 100 epochs and report mean and std on 3 runs in tab. 7. We note that the baseline is the superior architecture among the trained architecture and

Nevertheless, using our method with M=2, we were able to recover the default architecture. We hypothesise that the reason for default architecture having the best performance is that current ResNet architecture is highly optimized for ImageNet dataset.

