# OpenReview forum: "Balanced Mixture of Supernets for Learning the CNN Pooling Architecture"
_automl.cc/AutoML/2023/Conference — AutoML 2023 MainTrack_

### Official Review · Reviewer_cD7Q · 2023-03-28

**Potential Impact On The Field Of Automl Rating:** 2
**Technical Quality And Correctness:** In technical terms, the paper appears…
**Technical Quality And Correctness Rating:** 4
**Clarity Rating:** 3
**Actions Required To Increase Overall Recommendation:** Please see the comments above.

**Summary Of Contributions:**

This paper discusses the importance of downsampling layers in CNNs and their impact on network performance. The study shows that predefined downsampling configurations are not optimal and proposes a new approach of a balanced mixture of SuperNets to optimize pooling configurations. The proposed method outperforms other methods and improves the default pooling configurations in CIFAR10, CIFAR100, and Food101 datasets.

**Clarity:**

It would be beneficial if the author included a table to clarify the design of the search space in the paper. The reason for having N positions for pooling in the search space is not entirely clear in different sections of the paper.

**Overall Review:**

1. The method only achieves marginal improvement compared to the previous method, which is a very important baseline, DiffStride.
2. The paper lacks a report on the efficiency of the proposed method, such as the number of GPU hours required for the experiments.
3. The experiments in the paper were limited to CIFAR-level and food-101 datasets, which are relatively small. It is recommended that the author conduct experiments on larger datasets, such as ImageNet, to demonstrate the potential of the proposed method.

**Potential Impact On The Field Of Automl:**

The proposed approach, specifically for CNNs, can potentially impact the field of AutoML by improving the effectiveness of network architecture search for image classification tasks. The method reduces the weight-sharing and inter-influence of pooling configurations, allowing for more accurate and efficient optimization of network architectures.

**Reproducibility (Optional):**

The author has made the code available at https://anon-github.automl.cc/r/Multi-Model-NAS-264B, which appears to be reliable based on the information provided in the paper. However, I have not personally verified the accuracy of the code.

**Review Confidence:**

4: You are confident in your assessment, but not absolutely certain. It is unlikely, but not impossible, that you did not understand some parts of the submission or that you are unfamiliar with some pieces of related work.

**Review Rating:**

9: Strong Accept: Technically flawless paper with major impact and strong evaluation, with no obvious flaws. Should be highlighted in the program.

**Review Summary:**

Please refer to the comments above. Considering the objective limitations, I am inclined to recommend a borderline leaning reject.

---

> ### Author Response · Authors · 2023-05-01
> **Response to Reviewer cD7Q**
>
> We thank the reviewer for their valuable comments and suggestions. We noted from the response that, whilst the review recommended strong accept, the reviewer may have intended to recommend borderline rejection. We hope our clarifications will help to clarify the review concerns. We address the concerns and answer the questions below:
>
> ### Clarity and Presentation
>
> > "It would be beneficial if the author included a table to clarify the design of the search space in the paper. The reason for having N positions for pooling in the search space is not entirely clear in different sections of the paper."
>
> In supplementary material, we added table 5 that clarifies the search space design by providing detail about the number of pooling operations and layers which their placements are searched. We fixed the number of pooling operations as the default backbone architectures and applied a few restrictions (no manipulation of input image, smooth and monotonic changes in feature map size vs. layers) to where pooling is allowed to be placed.
>
>
> ### Comparison with baselines
>
> >"The method only achieves marginal improvement compared to the previous method, which is a very important baseline, DiffStride."
>
> Diffstride search space is continuous, which allows for fractional pooling allowing for kernels of different size and shapes (not necessarily square). However, it requires replacing pooling layers with a downsampling module that involves complex number computation. Our method achieves similar results within a discrete search space, with minimal change to the backbone overall architecture.
>
> ### Training Time
>
> >"The paper lacks a report on the efficiency of the proposed method, such as the number of GPU hours required for the experiments."
>
> We updated table 2 to include GPU hours for each method.
>
>
> ### Additional Experiments
>
> >"The experiments in the paper were limited to CIFAR-level and food-101 datasets, which are relatively small. It is recommended that the author conduct experiments on larger datasets, such as ImageNet, to demonstrate the potential of the proposed method."
>
> After submission, we evaluated our approach also on ImageNet. We included the results of our tests in table 7. We believe that the pooling configurations for ResNet models are optimised on ImageNet and therefore our identified pooling configurations are near optimal, and more importantly, our method is able to recover the default configuration (for M=2 SuperNets).

---

### Review · Reproducibility_Reviewer_ckpj · 2023-04-12

**Completeness Of Code And Dataset Supplement Rating:** 3
**Usability And Ease Of Reproducibility Rating:** 1

**Actions Required To Increase The Reproducibility And Overall Recommendation:**

The recommendation to the authors is to start with a clean repository and new virtual environment and follow their instructions. This will allow the authors to fix the problems with the provided requirements.txt file. When the instructions and provided requirements.txt lead to a working environment, this will allow for further reproducibility review.
Furthermore, to prevent any more issues, it would be wise to check whether all instructions in the given README.md files work and give the correct results in this environment.

**Completeness Of Code And Dataset Supplement:**

The provided instructions and requirements.txt are insufficient to reproduce the virtual environment necessary to run the code. Therefore, in its current state, it is not possible to verify further reproducibility of the code and dataset.

ffcv throws an error on installation:
File "/tmp/pip-install-skbf3z9b/ffcv/setup.py", line 30, in <module>
        extension_kwargs = pkgconfig('opencv4', extension_kwargs)
A potential solution is to follow instructions which are not included with the code but can be found at https://pypi.org/project/ffcv/#install-with-anaconda:
conda create -y -n ffcv python=3.9 cupy pkg-config compilers libjpeg-turbo opencv pytorch torchvision cudatoolkit=11.3 numba -c pytorch -c conda-forge
However, if the overlapping packages are not removed from requirements.txt, this leads to errors.
If they are removed, the following error appears: ERROR: Failed building wheel for Pillow_SIMD
due to /usr/include/limits.h:26:10: fatal error: bits/libc-header-start.h: No such file or directory

**Overall Reproducibility Review:**

The authors structured the instructions for reproducing the results in the paper into different folders, which makers reproducing easier.
However, no further feedback can be given to the paper's reproducibility, due to the issues with installing the required packages.

**Review Confidence:**

3: You are fairly confident in your assessment. It is possible that you did not understand some parts of the submission or that you are unfamiliar with some pieces of the code or data.

**Review Rating:**

2: Strong reject, the paper appears to be completely unreproducible.

**Review Summary:**

The paper's experiments are not reproducible because the provided instructions and requirements.txt file will not result in a working virtual environment.

**Summary Of Necessary Code And Dataset Supplement:**

The paper presents a novel method named "Balanced Mixtures" to estimate the optimal pooling types and locations in the residual network for a given dataset using a balanced mixture of SuperNets. The method is an extension to the single-path one-shot (SPOS) method originally introduced by (Guo et al., 2020; Li and Talwalkar, 2020; Stamoulis et al., 2020).
The ResNet-18, ResNet-20 and ResNet-50 architectures as proposed in He et al., 2016 are used. For all architectures, the fully connected layer and strided convolutions are replaced by max pooling operations.
The CIFAR-10, CIFAR-100 and Food101 datasets are used for evaluation.
NAS methods used for comparison are SPOS, Differentiable architecture search (DARTS) (Liu et al., 2018b), Monte-Carlo Tree Search (MCTS) (Su et al., 2021a) and Boltzmann Softmax Exploration (BSE) (Asadi and Littman, 2017; Cesa-Bianchi et al., 2017).
Non-NAS methods used for comparison are DiffStride (Riad et al., 2022), DynOPool (Jang et al., 2022) and Shape Adaptor (Liu et al., 2020).

**Usability And Ease Of Reproducibility:**

Please refer to Completeness Of Code And Dataset Supplement

---

> ### Author Response · Authors · 2023-05-01
> **Response to Reproducibility Reviewer ckpj**
>
> We thank the reviewer for their through feedback on our code.
> We updated the github directory with new installation instructions. Please note that FFCV library is not required for running experiments in resnet20 folder and only necessary for food101 and cifar folder.
>
> FFCV should be installed on the python 3.9 environment. Please note the torch version:
>
> ```
> >>> import torch
> >>>torch.__version__
> '1.13.1+cu117'
>
> ```
>
> We tested and verified installation by two methods:
>
> ```
> pip install -r requirements.txt
> ```
>
> Alternatively:
>
> ```
> conda create -n ffcv2 python==3.9
> conda activate ffcv2
> pip install -r requirements.txt
>
> ```
>
>
> Please note that FFCV might have problems for installation in older cuda versions.
> We also added approximate run time for each experiment in a table in the repository, the run time does not consider the data preparation, only the architecture search/training.

---

### Official Review · Reviewer_v6iq · 2023-04-13

**Potential Impact On The Field Of Automl Rating:** 4
**Technical Quality And Correctness:** The contributions seem correct to me.
**Technical Quality And Correctness Rating:** 4
**Clarity:** The paper is clearly written.
**Clarity Rating:** 4

**Summary Of Contributions:**

This paper studies the placement of layers that reduce spatial dimensionality in convolution-based neural architecture search (NAS) search spaces for the vision domain. These decisions are often treated as fixed in such search spaces, often with a predefined number of blocks before each spatial pooling location. The authors first observe that standard weight-sharing-based NAS techniques fail to recover this optimal placement, and use a ResNet-20 backbone on three vision datasets as a testbed for this problem – this setup admits a small search space of only 36 architectures and allows for convenient evaluation of the de facto choices made by standard vision backbones, which turn out to be suboptimal. The authors propose a new weight-sharing NAS method that targets this issue–balanced mixtures of supernets, which reduces the amount of weight-sharing between the different supernets. The authors evaluate their method on three vision datasets and show that their method improves performance by searching over different pooling placement choices alone.

**Actions Required To Increase Overall Recommendation:**

- Additional backbone evaluations
- Timing comparisons
- Additional datasets (lower priority)


**Overall Review:**

Strengths:
- The authors have identified an important problem in the NAS literature that appears to have been mostly understudied by prior work.
- The balanced mixture of supernets idea appears to work for this problem – it is also interesting that prior weight-sharing methods fail.
- A ResNet-based search space for studying this problem is extremely simple and seems to be exactly the right place to start for this problem.

Weaknesses:
- It would be great to include an evaluation of other search spaces besides the ResNet-based search space in order to show that the results hold up when using other blocks. However, I do not view this as a deal breaker because the current results are interesting on their own.
- One thing that seems missing is that since the proposed method is no longer single-shot in the usual weight-sharing sense, it could make sense to report training times and parameter counts compared to weight-sharing methods. While this is understandably not the point of the work, this is useful information.
- Additional datasets would be nice to have, since the evaluation comprises only three well-known vision datasets.

Other comments:
- Another interesting point that this work points out is the limitations of weight-sharing-based NAS. It would be interesting to study these limitations further.
- Lines 340-341: “One could argue that NAS in general are a waste of computation, however, they help to avoid an even more expensive validation search for the optimal hyper-parameters of the model” – I’m not sure I agree with this, can the authors comment on this view?


**Potential Impact On The Field Of Automl:**

NAS is a core topic within AutoML, and this work has a strong potential for impact within NAS.

**Review Confidence:**

3: You are fairly confident in your assessment. It is possible that you did not understand some parts of the submission or that you are unfamiliar with some pieces of related work.

**Review Rating:**

7: Weak Accept: Technically sound paper with moderate-to-high impact and strong evaluation, with perhaps some minor flaws.

**Review Summary:**

The problem that the authors have identified is important, and their experimental design seems very appropriate. The resulting method also seems to work, while prior work fails on this simple search space, which is interesting. The evaluation could be strengthened by extending to additional backbones beyond ResNet, a comparison of search timings to standard weight-sharing methods, and additional datasets.

---

> ### Author Response · Authors · 2023-05-01
> **Response to Reviewer v6iq**
>
> We thank the reviewer for their valuable comments and suggestions. We address the concerns and answer the questions below:
>
> ### Additional Experiments
>
> > "Additional datasets would be nice to have, since the evaluation comprises only three well-known vision datasets."
>
> > "Actions Required To Increase Overall Recommendation: Additional datasets (lower priority)"
>
> After submission, we evaluated our approach also on ImageNet and report results on ImageNet in table 7 in the appendix. In that case, the algorithm does not improve over the initial pooling configuration, we believe that neural network configurations may have been heavily optimized manually for the ImageNet data. Nevertheless, we are also able to show the advantage of using a Mixture of Supernets instead of just one. In fact, MoS manages to find the optimal configuration when only using two models, while one model is unable to.
>
>
>
> > "It would be great to include an evaluation of other search spaces besides the ResNet-based search space in order to show that the results hold up when using other blocks. However, I do not view this as a deal breaker because the current results are interesting on their own."
>
> Unfortunately, the limited time for rebuttal did not allow us to perform experiments on additional backbone architectures, however it is possible to potentially add these results in the camera ready version if the paper is accepted.
>
>
> ### Training Time and Parameter Count
>
> > "it could make sense to report training times and parameter counts compared to weight-sharing methods."
>
> > "Actions Required To Increase Overall Recommendation: Timing comparisons"
>
> We updated table 2 to include search times for each method. The number of parameters in experiments presented on table 2 are the same as default ResNet, except for our method which has M times number of parameters than default configuration during the search, but the found model will still have the same amount of parameters as a default ResNet.
>
> ### Other Comments
>
> > "Lines 340-341: “One could argue that NAS in general are a waste of computation, however, they help to avoid an even more expensive validation search for the optimal hyper-parameters of the model” – I’m not sure I agree with this, can the authors comment on this view?"
>
> We consider weight sharing as a trade-off between the amount of computation needed to find the optimal solution and the quality of an approximate solution. The extreme case is brute-force in which there is no weight sharing and each model is trained independently but the found solution is optimal. The other extreme is to use a single SuperNet model, in which the training is fast, but the found solution is biased by the interference of each model. Our Mixture of SuperNets is a compromise in which we reduce the interference by using multiple SuperNets, but increase the training time.

---

> > ### Comment · Reviewer_v6iq · 2023-05-09
> > **Thank you for your response**
> >
> > Thank you for your response -- I have updated my score from a 7 to an 8 for including an additional dataset and timing results.

---

### Official Review · Reviewer_qZsK · 2023-04-13

**Potential Impact On The Field Of Automl Rating:** 4
**Technical Quality And Correctness Rating:** 3
**Clarity Rating:** 3

**Summary Of Contributions:**

The authors explore the impact of downsampling layers on the performance of CNNs and highlight the limitations of current NAS methods for optimizing downsampling configurations. The authors propose a balanced mixture of SuperNets to address these limitations and show that this approach outperforms other approaches and default pooling configurations on CIFAR10, CIFAR100, and Food101 datasets. The authors emphasize the importance of appropriate search space design and present a solution for learning the optimal scale of CNN feature maps through pooling/stride configurations. The proposed method reduces weight-sharing and inter-influence issues associated with pooling configurations, which leads to better ranking correlations. Finally, the authors also compare their proposed method with non-NAS-based approaches.

**Actions Required To Increase Overall Recommendation:**

The authors should enhance the paper by adding more experimental results in terms of datasets and architectures.

The accuracy aspect is fine, but the authors should also include FLOPs or the number of multiplication and accumulation (MAC) operations. Late pooling has better accuracy compared to early pooling, as shown in Figure 1. However, networks with early pooling have lower FLOPs and MACs due to a reduction in spatial resolution early in the network. The subsequent layers operate on lower spatial resolution feature maps. From Table 3, the baseline methods seem to work better than the proposed method in this paper. Therefore, FLOPs and MACs can be good for enhancing the paper.

Minor:
Line 135: r^L, L (number of layers) is not defined before. It should be clarified.

Line 136-137: therefore it is reasonable to exclude upsampling operations from the search space.: This statement seems to be obvious and redundant. The authors established already that they are searching for optimal downsampling operations.

The reference on line 656 is missing.

**Clarity:**

First of all, the reviewer enjoyed reading the article. The reviewer mainly appreciates the step-by-step approach and analysis. The figures support the proposed search space and algorithm. The writing is clear and the technical contributions are novel.

**Overall Review:**

The authors should clarify if this is the first kind of work to use NAS for the position of downsampling layers. The authors should probably take a look at the following paper and consider explaining the difference between their work: https://ieeexplore.ieee.org/document/9506777

It may be a good idea to clarify in the search space subsection how the authors obtained 36 configurations for Resnet20 instead of placing it in Section 3.1.

The GPU search time is not reported, and hence, the authors should also compare the computation time. The time required to train 36 architectures vs the search time of the proposed algorithm would be good to present in the paper. This can signify the importance of the search method.

The experiments are limited. The results on large-scale datasets and resolution are missing. At least, the authors should have evaluated the mixture of supernets on the Imgenet-16-120 dataset. The authors experimented only on the Image Classification task and CNN-based network. Currently, Vision transformers are being utilized on various tasks and slowly replacing CNNs. Hence, the proposed search method should be tested on different kinds of architecture and tasks to truly show its effectiveness.



**Potential Impact On The Field Of Automl:**

The chosen search space is interesting and presents exciting results. The mixture of experts (MoE) approach itself is quite interesting and presents new insights. It can be utilized to solve other kinds of search problems. The paper provides new directions in the field of NAS.

**Review Confidence:**

4: You are confident in your assessment, but not absolutely certain. It is unlikely, but not impossible, that you did not understand some parts of the submission or that you are unfamiliar with some pieces of related work.

**Review Rating:**

7: Weak Accept: Technically sound paper with moderate-to-high impact and strong evaluation, with perhaps some minor flaws.

**Review Summary:**

The presentation of the proposed approach is good. However, more evaluation is required to justify the correctness of the paper. The reviewer is fairly confident of the review after having gone through the search space and search algorithm. The review covered essential aspects of the paper, including the novelty and evaluation of the proposed approach.

**Technical Quality And Correctness:**

The search algorithm detailed in Sections 2.2 and 2.3 is solid and is explained very well. The mixture of SuperNets is technically sound and the authors did a great job of explaining it. The problem of automatically finding the position of downsampling layers presents a new direction.

---

> ### Author Response · Authors · 2023-05-01
> **Response to Reviewer qZsK**
>
> We thank the reviewer for their valuable comments and suggestions. We address the concerns and answer the questions below:
>
> ### Search Space Design
>
> > "It may be a good idea to clarify in the search space subsection how the authors obtained 36 configurations for Resnet20 instead of placing it in Section 3.1."
>
> We agree. We also added table 5 in supplementary material to clarify the design of search space in all experiments.
>
> ### Search Time
>
> > "The GPU search time is not reported."
>
> We included the search time in table 2.
>
> ### Additional Experiments
>
> > "The experiments are limited. The results on large-scale datasets and resolution are missing."
>
> After submission, we evaluated our approach also on ImageNet. We included the results of our tests on table 7. In that case the algorithm does not improve over the initial pooling configuration, we believe that neural network configurations may have been heavily optimized manually for the ImageNet data. Nevertheless, we are also able to show the advantage of using a Mixture of Supernets instead of just one. In fact, MoS manages to find the optimal configuration when only using two models, while one model is unable to.
>
> ### Other Concerns
>
> > "the authors should probably take a look at the following paper and consider explaining the difference between their work: https://ieeexplore.ieee.org/document/9506777
> "
>
> In this paper authors try to optimize the pooling of a U-Net model by choosing among three different operations that change the position of the pooling layer. While this is an interesting way to optimize the pooling layers, it is a local optimization that greatly reduces the search space. For instance, using that trick and assuming that it works well (in our extensive experiments DARTS based approaches do not work for pooling), we believe we would not be able to find the optimal configuration on FOOD-101 because the pooling configurations are further than 3 steps from the default setting.
>
> > "Currently, Vision transformers are being utilized on various tasks and slowly replacing CNNs. Hence, the proposed search method should be tested on different kinds of architecture and tasks to truly show its effectiveness."
>
> As ViT does not use downsampling layers, it is not possible to use our method on it. Swin [1] instead proposes a kind of analysis at multiple resolutions, and there our method could be used. However, we do not have the time and the computational resources to obtain results for the rebuttal.
>
> *[1] Liu, Ze, et al. "Swin transformer: Hierarchical vision transformer using shifted windows." Proceedings of the IEEE/CVF international conference on computer vision. 2021.*
>
> ### Corrections
>
> >"Minor: Line 135: r^L, L (number of layers) is not defined before. It should be clarified."
>
> > "Line 136-137"
>
> > "The reference on line 656 is missing."
>
> Corrected.

---

> > ### Comment · Reviewer_qZsK · 2023-05-08
> > **Reply to Authors**
> >
> > The reviewer would like to thank the authors for their response and incorporating the changes in the paper based on the feedback provided. However, for a few points raised by the reviewer, the authors did not write a rebuttal. The ImageNet experiments are required to strengthen this work and the authors did provide the results in Table 7. However, the results show accuracy compared to the baseline. However, as suggested before, the authors should have used FLOPs or MACs as a metric to compare which can boost the significance of this work. The authors are yet to provide results in this direction. The reviewer agrees with the authors claim “we believe that neural network configurations may have been heavily optimized manually for the ImageNet data.” The reviewer’s rating is still 7 (weak accept) as the comparison of networks using other metrics are still missing in the updated version.

---

> > > ### Author Response · Authors · 2023-05-16
> > > **Response to Reviewer**
> > >
> > > Thank you very much for feedback on our response.
> > >
> > > To investigate the computational cost of our method compared to other works, we measured the relative FLOPs with respect to the baselines for Diffstride as 1.8 compared to our method with 2.22. While the FLOPS are lower in their case, the training time is similar with 1.43 (Diffstride) vs. 1.5 (ours) with respect to baseline training time. However, our method works with discrete and square filters which remains compatible with basic building blocks with minimal change to the backbone architecture while Diffstride search space is continuous, which allows for fractional pooling allowing for rectangular kernels.
> > >
> > > Furthermore, the aim of this work is to find pooling configurations that improve the accuracy of a classification model (which we show is a hard task by itself) and our method does not apply any constraints on computational cost, and as mentioned in the review, the resulting late pooling architectures have higher FLOPs. At the moment we do not have a simple way to optimize pooling while constraining the FLOPs. Instead, we could optimize a multi-cost objective that takes into account also the computational cost of the found models. However, this is complex and goes beyond the scope of our current work.

---

### Official Review · Reviewer_Zu75 · 2023-04-18

**Potential Impact On The Field Of Automl Rating:** 2
**Technical Quality And Correctness:** 1. This paper claimed, "In this work …
**Technical Quality And Correctness Rating:** 2
**Clarity:** This paper is readable and well written.
**Clarity Rating:** 3
**Actions Required To Increase Overall Recommendation:** Please try to address my major concer…

**Summary Of Contributions:**

This paper found the existing one-shot NAS algorithms cannot be natively applied to find optimal pooling configuration. This paper hypothesized that the existing weight sharing results in this problem. So, a mixture of supernets is proposed. They use multiple supernets rather than a single supernet to estimate the accuracy of a pooling configuration. The proposed algorithm is demonstrated on CIFAR datasets and Food101.

**Overall Review:**

This paper noted the configuration of pooling layers will affect performance. A new one-shot search algorithm is proposed to search for pooling layer configuration. Multiple supernets are used to estimate the performance of sub-net.

As I mentioned, this paper has problems with experiments and hypotheses. I agree the configuration of pooling will affect the final performance to some extend. I am not sure 1) the proposed algorithm can effectively outperform baselines and 2) that similar experimental result will be observed on large-scale datasets and harder tasks.

**Potential Impact On The Field Of Automl:**

Previous popular search spaces always have downsampling as a candidate operator. This paper actually searches for the macro architecture, i.e. placement of downsampling layers. Their hypothesis and experimental results do not make sense to me. I do not expect a lot of discussions of this paper.

**Review Confidence:**

4: You are confident in your assessment, but not absolutely certain. It is unlikely, but not impossible, that you did not understand some parts of the submission or that you are unfamiliar with some pieces of related work.

**Review Rating:**

3: Reject: For instance, a paper with technical flaws, weak impact, and/or weak evaluation.

**Review Summary:**

The hypothesis and experimental results are not solid and convincing. The algorithm contribution is weak.

---

> ### Author Response · Authors · 2023-05-02
> **Response to Reviewer Zu75 (part 1/2)**
>
> We thank the reviewer for their comments and suggestions. We address the concerns and answer the questions below:
>
> ### Question 1
>
> >  " 1. This paper claimed, "In this work we tested both differentiable and sample based approaches, but both failed to provide good results for finding the optimal pooling configuration of a network. We hypothesize that the underlying reason is two fold: inappropriate search space design, and strong weight sharing in SuperNet." "
>
> We agree that maxpooling operations are included in many search spaces, however, they do not contribute to the feature map size of the layer, for example in popular DARTS [1] search space, the downsampling of the feature map is performed by manually setting the strides of operations adjacent to the input nodes equal to two, the stride of all other operations (including the maxpoolings are set to one).
>
> Some previous work such as DenseNAS [1][add more] do not use downsampling directly into the search space, but they consider configurations with skip operations which can induce a reduction of the number of layers at a given resolution by reducing the total length of the network. Thus, in those search spaces it is not really possible to define configurations with similar length but different pooling configurations, which is what we consider.
> Additionally, those papers mixed skip operations with other layers’ operations such that it is not possible to discern if an improvement in performance is due to a different global pooling configuration or different operations used also because the optimal configuration is in most of the cases not available.
>
> Finally, most work in recent years focuses on the micro-structure of CNN. Commonly used search spaces such as DARTS-based search space [2] and Mobilenet-based search space [3], and benchmarks such as NAS-Bench-201 [4] and NAS-Bench-Macro [5] all fix the outer-skeleton of the network.
>
> *[1]  Liu, Hanxiao, Karen Simonyan, and Yiming Yang. "Darts: Differentiable architecture search." arXiv preprint arXiv:1806.09055 (2018). \
> [2]  Fang, Jiemin, et al. "Densely connected search space for more flexible neural architecture search." Proceedings of the IEEE/CVF conference on computer vision and pattern recognition. 2020.  \
> [3] Sandler, Mark, et al. "Mobilenetv2: Inverted residuals and linear bottlenecks." Proceedings of the IEEE conference on computer vision and pattern recognition. 2018.  \
> [4] Dong, Xuanyi, and Yi Yang. "Nas-bench-201: Extending the scope of reproducible neural architecture search." arXiv preprint arXiv:2001.00326 (2020). \
> [5] Su, Xiu, et al. "Prioritized architecture sampling with monto-carlo tree search." Proceedings of the IEEE/CVF Conference on Computer Vision and Pattern Recognition. 2021.*
>
> ### Question 2
>
> > " Why did this toy example motivate you to propose a mixture of models?"
>
> As studied previously [1,2] the coupling of architectures in a supernet training can have a negative impact on the evaluation performance of the architecture. This simple example shows that when training multiple (in this case two) pooling configurations with the same weights produces interference and leads to lower performance than training models independently. The performance when mixing the two pooling configurations during training goes from 91.83 to 89.84 for the late pooling and from 87.45 to 86.81 for early pooling). Thus, it motivates our approach of reducing this interference by using multiple specialised models (SuperNets) in the same training.
>
> *[1] Bender, Gabriel, et al. "Understanding and simplifying one-shot architecture search." International conference on machine learning. PMLR, 2018.  \
> [2] Zhang, Yuge, et al. "Deeper insights into weight sharing in neural architecture search." arXiv preprint arXiv:2001.01431 (2020).*
>
> > "Furthermore, the searched result shows accuracy 91.55 in Table 2 that is still inferior than late pooling."
>
> The late pooling configuration is better than the found configuration, but it is not found automatically, without brute force. Among all tested methods, only our approach managed to get close to the optimal pooling configuration, which shows the difficulty of the task.
>
> ### Question 3
>
> > "The experiments are not solid.... Did you carefully set hyperparameters or optimizers? To my knowledge, ResNet20 usually reports around 91.x% on CIFAR-10. In my opinion, baselines are not well trained."
>
> The hyperparameters of the proposed baselines are carefully tuned. As explained in Appendix B.2, the architecture we use is a modified ResNet20, with fewer parameters, the last fully connected layer is removed and the channels in the last layer are lower. This produces lower baselines than those mentioned by the reviewer, but allows us to conduct the brute force experiment for the optimal pooling configuration more efficiently.

---

> > ### Author Response · Authors · 2023-05-02
> > **Response to Reviewer Zu75 (part 2/2)**
> >
> > > "On the other hand, Late Pooling (91.83%) is still better than the Balanced Mixture (91.55%)"
> >
> > The late pooling configuration is better than the found configuration, but it is not found automatically, only via a bruteforce baseline search. However, among all tested methods, only our approach managed to get close to the optimal pooling configuration, which shows the difficulty of the task. Late pooling configuration in Figure 1, is chosen as an example to show the effect of weight sharing on architecture performances. Since the architecture is not found by NAS, the performance of this specific configuration is available from bruteforcefully testing all possible configurations, in table 5 of appendix D.1. Not all late pooling architectures are performing well, therefore before performing bruteforce training it’s not possible to choose an appropriate pooling configuration.
> >
> > ### Question 4
> >
> > > " Because the experimental result is not solid on small scale CIFAR datasets. I cannot expect the proposed algorithm works on large-scale datasets, like ImageNet."
> >
> > We hope that the previous answers have clarified and demonstrated that our evaluation is solid and fair and that our approach outperforms several well-known and common NAS and non-NAS algorithms for this problem.
> > Furthermore, we also included results on ImageNet in table 7 in appendix. In that case, the algorithm does not improve over the initial pooling configuration, however it seems that neural network configurations may have been heavily optimized manually for the ImageNet data. Nevertheless, we are also able to show the advantage of using a Mixture of Supernets instead of just one. In fact, Mixture of Supernets manages to find the optimal configuration when only using two models, while one model is unable to find it.
> >
> > > "I guess 1) we cannot observe difference between different pooling configuration "
> >
> > We showed with our benchmark that pooling can highly affect the final performance of a CNN model. For each pooling configuration we also provided standard deviation over 3 runs, which shows that the provided results are significant.
> >
> > > "2) it is very hard to outperform the stander configuration by changing pooling layers."
> >
> > We agree with the reviewer that outperforming the standard pooling configurations is a significant challenge, particularly for datasets and architectures that have been heavily optimized manually. This is why we consider our work and contribution to be important.
> >
> > ### Response to Overall Review and Summary
> >
> > > " I am not sure 1) the proposed algorithm can effectively outperform baselines "
> >
> > We hope that our modifications have clarified the motivation for investigating pooling architectures and results on datasets including ImageNet. We show that for datasets different from ImageNet our algorithm can find configurations that are better than the common baseline.
> >
> > > "2) that similar experimental result will be observed on large-scale datasets and harder tasks."
> >
> > We have demonstrated that our algorithm can recover the baseline configuration for ImageNet, however with no further improvement. We believe that the ResNet pooling configuration is already near-optimal for this ubiquitous dataset, due to heavy manual optimization.
> >
> > > "The hypothesis and experimental results are not solid and convincing. The algorithm contribution is weak."
> >
> > We hope our responses have clarified our hypotheses, motivation and experimental results. Finding a way to deal with the problems of weight sharing and the exponential number of models is an important issue in AutoML and here we propose and verify a novel approach based on Mixtures of Supernets.

---

### Official Review · Reviewer_xdDU · 2023-04-18

**Potential Impact On The Field Of Automl Rating:** 3
**Technical Quality And Correctness Rating:** 3
**Clarity Rating:** 3
**Actions Required To Increase Overall Recommendation:** 1.	The work has studied the search fo…

**Summary Of Contributions:**

This paper focuses on downsampling layers, such as pooling and strided convolutions, in convolutional neural networks (CNNs). The authors analyze the performance of independently trained models with different pooling configurations on CIFAR10 using a ResNet20 network. They find that the position of downsampling layers significantly impacts network performance and that predefined downsampling configurations are not optimal. They also show that traditional one-shot Network Architecture Search (NAS) based on a single SuperNet does not work for optimizing pooling configurations due to parameter sharing issues. To address this, they propose a balanced mixture of SuperNets that associate pooling configurations with different weight models, reducing weight-sharing and inter-influence of pooling configurations on the SuperNet parameters. Their proposed approach outperforms other methods and improves over default pooling configurations on CIFAR10, CIFAR100, and Food101 datasets.

**Clarity:**

In general, the paper is well-written with illustrative figures. However, some symbols should be defined first. And the description of the search space should be elaborated.

**Overall Review:**

Paper strength:
1.	The motivation of the paper is interesting and clear.

Paper weakness:
1.	The presentation of this paper is not good and some details are not explained clearly.
2.	The experiments are not convincing.
3.	More in-depth analysis is expected. It is not enough to show comparable / better results than existing methods.


**Potential Impact On The Field Of Automl:**

This paper presents an interesting angle to NAS from the perspective of pooling operations. However, whether the claims or findings can scale beyond small-scale datasets (CIFARs) and other types of architectures are yet to be verified.

**Review Confidence:**

3: You are fairly confident in your assessment. It is possible that you did not understand some parts of the submission or that you are unfamiliar with some pieces of related work.

**Review Rating:**

7: Weak Accept: Technically sound paper with moderate-to-high impact and strong evaluation, with perhaps some minor flaws.

**Review Summary:**

Overall, this research contributes to the understanding of the impact of downsampling layers in CNNs and presents a novel approach for optimizing pooling configurations, which could have practical implications in the field of deep learning.

**Technical Quality And Correctness:**

The proposal of using a balanced mixture of SuperNets to address the parameter sharing issue in traditional one-shot Network Architecture Search (NAS) is innovative and shows promising results. The paper is well-written and provides clear insights into the limitations of existing approaches and the proposed solution.

---

> ### Author Response · Authors · 2023-05-01
> **Response to Reviewer xdDU**
>
> We thank the reviewer for their valuable comments and suggestions. We address the concerns and answer the questions below:
>
> ### Experimental Results
>
> > "whether the claims or findings can scale beyond small-scale datasets (CIFARs) and other types of architectures are yet to be verified."
>
> After submission, we evaluated our approach also on ImageNet (table 7 in appendix). We could not find better pooling configurations than the default. We believe that the pooling configurations for ResNet models are optimised on ImageNet and therefore our identified pooling configurations are near optimal, and more importantly, our method is able to recover the default configuration (for M=2 SuperNets).
>
> > "2. The experiments are not convincing. 3. More in-depth analysis is expected. It is not enough to show comparable / better results than existing methods."
>
> We show that finding the pooling configuration is hard even with a limited set of pooling configurations, and existing methods fail. We present a method that works and compare it with NAS and not NAS approaches, and in all cases results are comparable or better.
>
>
> ### Clarity and Presentation
>
> > "some symbols should be defined first. And the description of the search space should be elaborated."
>
> > "The presentation of this paper is not good and some details are not explained clearly. "
>
> We added the missing definitions in several places in text in page 3-4 of the revised version of the paper and we also improved the description of the search space and added table 5 in the supplementary material.
>
> ### Recommended Actions
>
> > "The work has studied the search for the optimal network under a fixed number of parameters. How about when you fixed the FLOPs?"
>
> The aim of this work is to find pooling configurations that improve the accuracy of a classification model, which we show is an unsolved and hard task by itself. At the moment we do not have a simple way to optimise pooling while keeping fixed the FLOPs. Instead, we could optimise a multi-cost objective that takes into account also the computational cost of the found models. However, this is complex and goes beyond the scope of our current work.
>
> >"The transferability of the found network should be studied. At least, networks found over CIFAR10 should be evaluated on other datasets."
>
> We do not claim that the found models are transferable, instead we claim that in order to obtain the best performance of a model, the pooling configuration should be optimised for the specific dataset; we show that common approaches do not work and propose an efficient way (compared to brute force) to do that.

---

### Author Response · Authors · 2023-05-01
**Overall Response:**

We thank the reviewers for their detailed reading of the manuscript and valuable comments. We are pleased that reviewers found our paper innovative, well-written with illustrative figures, interesting and with clear motivation and new directions in the field of NAS. We are also glad that our experiments are shown to be promising and exciting to the reviewers.
The main criticism from one reviewer is the possible previous work already providing results on the topic and the lack of solid experiments. We clearly explain that no previous work provided analysis/results on our specific topic and our results are solid, in line with baselines, with the correct hyper-parameters, and run multiple times.
Overall in the paper, we added results on ImageNet and we included time for training our model as asked by reviewers.

---

> ### Author Response · Authors · 2023-05-16
> **Author Rebuttal Discussion Reminder**
>
> Dear Reviewers,
>
> We would greatly appreciate it if you could review the updated version of the paper and responses to your comments and provide us with your questions and suggestions. Doing so will allow us to address additional questions you may have before the discussion phase ends.
> We again thank you for your time and effort reviewing our paper and look forward to your feedback.

---

### Comment · Program_Chairs · 2023-05-16
**Additional reproducibility review**

I only have a Windows machine with a GPU available and the environment is not made for that (Windows Docker cannot support GPU either sadly, AFAIK). Specifically, some of the requirements will never work with Windows in the specified/required version (e.g. FFCV) and would require newer version.

The requirements file has some weird double requirements like specific Pillow and Pillow_SIMD and the repo's documentation could certainly use a bit more polish. For example, some links are broken, no minimal example, and very short instructions. Nevertheless, from going through the code, it seems to me that everything is present and someone from the field with the right machine is likely able to reproduce the results with medium additional effort. Moreover, the code itself and it CLI-based usage is well enough documented in the python code (not in the readme). In general, it is hard to reproduce work like this as it has a lot of OS requirements like CUDA, which cannot be expressed in a requirements.txt and are here only stated implicitly in the readme. A Docker file would have been ideal to use with NVIDIA container toolkit.

Judging from the reproducibility review and the response, the authors have put additional effort into making the repo better. And in this state (as described above) it is not great but also not bad. IMO, I would give the reproducibility score a weak or borderline accept (according to the rating's text). Moreover, all my polish-releated problems could be fixed for the camera ready version.